# An Analysis of Switching Behavior from Traditional Hospital Visit to E-Health Consultation

**DOI:** 10.3390/healthcare13151784

**Published:** 2025-07-23

**Authors:** Shyamkumar Sriram, Harshavarthini Mohandoss, Nithya Priya Sunder, Bhoomadevi Amirthalingam

**Affiliations:** 1Department of Rehabilitation and Health Services, University of North Texas, Denton, TX 76201, USA; shyamkumar.sriram@unt.edu; 2Faculty of Management Sciences, SRIHER (DU), Chennai 600116, Tamil Nadu, India; harshaharsha98099@gmail.com; 3Amity Institute of Public Health and Hospital Administration, Amity University, Noida 201301, Uttar Pradesh, India

**Keywords:** telemedicine, patient behavior, healthcare adoption, digital health, patient satisfaction

## Abstract

With the rapid digital transformation of healthcare services in India, this study investigates the factors influencing the behavioral shift from traditional hospital visits to e-health consultations. The primary **objective** was to analyze patient attitudes, satisfaction, and perceived barriers to adopting virtual healthcare, especially in urban and semi-urban settings. **Methods**: The methodology adopted in the study was descriptive, and a convenience sampling technique was used for data collection because the feasible times of the patients’ availabilities were taken into consideration for data collection. Both primary and secondary data were collected using questionnaires and literature. A sample size of 385 participants was used in this study. Various statistical tools, such as frequency, ANOVA, and Chi-square tests, were used to test the hypotheses. **Results**: It was observed from ANOVA and Chi-square tests that the factors for switching from traditional consultation to e-health services have a positive association. It was found that integrating data through influencing factors significantly (*p* < 0.001) improved decisions on e-health services. **Conclusion**: This study highlights the shift from in-person to e-health consultations driven by convenience, flexibility, and pandemic-related needs while acknowledging barriers such as digital literacy, infrastructure gaps, and trust issues. It recommends strategies, such as secure platforms, training, and integrated care models, for a more inclusive digital health future.

## 1. Introduction

The global healthcare landscape is undergoing a profound transformation, marked by a rapid shift from traditional in-person medical consultations to digital health platforms. This transition, often referred to as the digitalization of healthcare, has been catalyzed by several interrelated factors: the increasing penetration of smartphones and the internet [1], advancements in artificial intelligence and telecommunication technologies [1], supportive governmental policies [2], and, most notably, the global COVID-19 pandemic [3]. During the pandemic, digital health emerged not as a convenience but as a necessity, leading to a surge in the adoption of teleconsultation platforms [4], mobile health (mHealth) applications, and virtual care delivery systems [5]. The concept of unified theory of acceptance and use of technology can be applied in information systems to design and observe usage behaviors that aims to provide a comprehensive framework [6], and the concept of the technology acceptance model (TAM) can be used to design new technologies based on perceived usefulness and perceived ease of use [7].

This shift has substantial implications in India, where the population is both vast and socioeconomically diverse. Traditionally, healthcare delivery has been centered on in-person interactions in clinics, hospitals, and community health centers. However, the pandemic disrupted this model by limiting physical access to medical services and simultaneously accelerating the demand for virtual consultations [8].

Urban and semi-urban populations, in particular, have shown increased receptivity to e-health platforms, owing to better access to technology, greater health awareness, and time constraints that make hospital visits less desirable [9]. By contrast, rural regions continue to face digital divides, limiting equitable access to these services [9].

The behavioral shift toward e-health is shaped by a mix of convenience- and necessity-driven motivations. From a patient’s perspective, e-health consultations offer numerous advantages, including reduced waiting times, cost savings, flexible scheduling, and, in many cases, the ability to consult specialists across geographic boundaries [10,11]. Digital platforms are increasingly perceived as viable alternatives to hospital visits, if not preferable, for routine follow-ups, prescription renewals, mental health support, and chronic disease management. Nevertheless, this transition is challenging. Trust deficits, concerns about the accuracy of virtual diagnostics, fear of privacy breaches, and a general lack of digital literacy remain significant barriers to wider adoption [12,13].

Socioeconomic and demographic variables also influenced this behavioral shift. Age, education level, income, occupation, and digital competence are closely linked to willingness and ability to use e-health services [14]. Older adults and individuals from low-income or less-educated backgrounds often exhibit reluctance or an inability to transition from the traditional care models. In contrast, younger tech-savvy users in metropolitan regions are more open to embracing digital platforms [15]. This disparity underscores the importance of understanding the determinants of switching behavior and addressing the contextual factors that shape patient choices. Institutional factors also play a role in shaping perceptions and trust. The lack of standardized protocols across different e-health providers, variable service quality, inconsistent user interfaces, and minimal regulatory oversight contribute to user hesitancy [16]. Strategic public health messaging, reliable digital infrastructure, and comprehensive training programs for both healthcare workers and patients are necessary to facilitate a seamless and sustainable transition.

This study employs the push–pull–mooring (PPM) framework as a theoretical lens to examine the behavioral shift from traditional hospital visits to e-health consultations. The PPM model, originally developed to explain migration patterns, has been adapted to understand consumer switching behavior in various contexts, including healthcare [17].

Therefore, this study sought to systematically examine the behavioral transition from hospital-based consultations to e-health platforms. It aims to identify and evaluate the key factors influencing patient decisions, such as service accessibility, digital convenience, and perceived value, while also addressing barriers such as digital trust and system usability [18]. By investigating these dimensions through a descriptive–analytical approach, this study offers insights that are not only academically valuable, but also practically applicable to healthcare planners, policy architects, and digital platform developers. Ultimately, it aims to support a more inclusive and efficient healthcare delivery ecosystem that aligns with the evolving needs and expectations of modern patients. The study highlights key factors influencing the shift to e-health, but lacks rural representation, provider perspectives, and qualitative insights. It overlooks long-term behavioral trends, digital literacy impacts, and platform comparisons. Privacy concerns, cultural factors, and trust have not been examined thoroughly. These gaps limit the broad applicability of the findings.

Therefore, the following objectives and Research Questions (RQ) were framed:RQ1. What are the key behavioral shifts that occur as patients transition from traditional hospital visits to e-health consultation?RQ2. What are the primary factors that influence patients’ adoption of e-health services?RQ3. What are the perceived advantages and challenges associated with e-health consultations compared to traditional in-person visits?RQ4. How does patient satisfaction and trust in e-health services compare to in-person consultations?

## 2. Materials and Methods

### 2.1. Literature Review

The “Determinants of Telemedicine Adoption Among Patients” found that accessibility was the strongest predictor of adoption, followed by usability and satisfaction. This patient-centered approach shifts the focus from technology to user experience. This highlights the importance of understanding how technology affects clinical workflows and patient outcomes, emphasizing the need for education beyond technical skills [19]. Telemedicine offers greater convenience and satisfaction for many patients but faces challenges such as technical reliability and limited physical examinations. This study calls for additional research and patient-focused improvement [20]. Service quality and payment models impact trust and risk perception, highlighting the need for consistent patient-centered care across digital and physical platforms. Studies have examined how social networks influence health information seeking among chronic patients, finding that younger, more interactive users prefer online sources, leading to greater self-efficacy and empowerment. This study highlights the need for healthcare providers to engage with online communities to guide informed decision-making. A study [21] highlighted that lockdown increased the use of telemedicine owing to its convenience, efficiency, and integration of e-health [9]. The research also emphasized the importance of several factors for effective and sustainable e-health adoption, such as a strong patient–doctor relationship for effective e-health consultation, a robust technical infrastructure, clear reimbursement policies, crucial need for data security and privacy, and the impact of employees’ willingness to embrace digital health solutions in the long term [22,23,24].

### 2.2. Participants

The study was conducted between January 2025 and March 2025 among the general population of Tamil Nadu. We developed a well-structured questionnaire to understand the behavioral shift from traditional hospital visits to health consultations. This is influenced by various factors such as age, digital literacy, and healthcare experience. A sample size of 385 was collected based on the population, and a sample size (Cochran) formula was used to calculate the sample for the study. The diversity within this sample allows for the generalization of the results to a larger population, providing a comprehensive understanding of the public’s willingness and readiness to adopt e-health consultations. This study was approved by the Internal Ethics Committee (IEC) of our institution. The limitation of the study was sample bias during data collection, which varied based on the type of respondents, as per their education and experience levels. There were a few potential biases in the online data collection, including the duplication of data and the accuracy of information provided in the study.

### 2.3. Study Design and Procedure

The research aimed to explore the descriptive study and the questionnaire developed using the COM B model; this study employed convenience sampling as it allows a quick and cost-effective assessment of participants with prior exposure to both traditional and e-health consultations. The questionnaire was aligned with the study’s aim to understand behavior patterns among early adopters of e-health services and was distributed to 50 sample respondents for a pilot study to validate the questionnaire using a Google form. Bias was minimized through several strategies, including ethical approval, diverse sampling across age, gender, and region, and the use of a structured, pilot-tested questionnaire. By clearly defining the inclusion and exclusion criteria, the participants were ensured that they had relevant experience with both traditional and e-health consultations. Statistical tools, such as ANOVA and Chi-square tests, were used to objectively analyze the associations.

The hypotheses mentioned in the study include

**H1****:** 
*There are no statistically collective efforts of the predictor (frequency of hospital visits, medical insurance coverage, multilingual support, waiting time reduction) on new usage of e-health services.*


**H2****:** 
*There is no association between exposure to illness in clinics and the recommendation of e-health services.*


The use of standardized Likert scales reduced measurement bias. Primary data provided multiple layers of insight into the study. By focusing on individuals across different socioeconomic strata and age groups, this study captured a broad and context-specific understanding of health care preferences within the city. Participants with varying degrees of exposure to digital healthcare solutions were included, enabling a well-rounded analysis of behavioral transitions. This combination of data sources contributed to the reliability and depth of the findings.

The inclusion criteria were as follows: (a) participants over 18 years of age had experience with both traditional and e-health consultations. (b) Studies or data focusing on telemedicine, behavioral psychology, or digital health adoption in India.

The exclusion Criteria were as follows: (a) participants who were not exposed to e-health services. (b) The literature focused solely on provider-side telemedicine implementation [19].

### 2.4. Questionnaire

The questionnaire had three components, including Section A, which consisted of demographic information, such as age, gender, education level (e.g., High school, Bachelor’s, or Master’s), and location (urban, semi-urban, or rural).

Section B consisted of a questionnaire on behavior shifts towards e-health consultations, including medical consultations, medical insurance for e-health consultations, multilingual support for e-health usage, waiting times, clinical motives, payment processes, health coaching and life science, rewards and discounts to motivate the employees on a 5-point Likert Scale (1 = Strongly Disagree, 2 = Disagree, 3 = Neutral, 4 = Agree, 5 = Strongly Agree).

Section C consisted of factors influencing the adoption of e-health services, including learning about e-health, decisions on e-health, benefits of e-health consultations, challenges faced during e-health consultations, a multifactor authentication process, and doctors’ transparency. All questionnaire items were developed using a 5-point Likert scale, indicating the behavior and acceptance of e-health services.

### 2.5. Statistical Analysis

Descriptive statistics were reported in this study. Frequency and percentage analyses were performed to determine the contribution of the e-health system. One-way ANOVA was used to compare the mean values of two or more independent groups to determine if there was a significant difference between the variables. In this study, illness at a clinic was an independent variable, recommendation of e-health services was the dependent variable, and trust in e-health services was an independent variable. Chi-square analysis was performed to observe the differences between independent and dependent variables. The association between demographic variables and exposure to illness at clinics, certification, and recommendation of e-health, payment processes, and multifactor authentication was analyzed. Statistical significance was set at *p* value < 0.05. Data were processed using R software version 1.0.0.

## 3. Results

A total of 385 of 400 respondents participated in this study. It was observed that the study used ANOVA, Chi-square analysis, and percentage analysis to analyze factors influencing switching behavior to the adoption of health services.

Table 1 shows the demographic data across four categories: age group, sex, education level, and location, using frequency and percentage values. The 26–35 age group had the highest representation of 142 respondents (37%), followed by the 18–25 age group, which had 125 (33%). Participation dropped significantly in older age groups, with those 60 and above accounting for just 11%. Females dominated the sample with 237 individuals (62%), while males comprised 148 individuals (38%), indicating a gender imbalance favoring females. Regarding education, 205 (53%) respondents had a bachelor’s degree, followed by 69 (18%), high school graduates made up 29%, while no participants reported having no formal education, showing a highly educated sample. Urban residents were the majority at 233 (61%), followed by semi-urban (75, 19%), and rural (77, 20%) participants, highlighting a strong urban bias. The overall data suggest that the sample primarily included young, urban, and well-educated females.

Table 2 shows the frequency and percentage of responses to the four healthcare service factors: medical insurance, multilingual support, reduced waiting times, and 24/7 availability. For medical insurance, the highest number of respondents (180, 46%) selected “Neutral,” indicating strongly disagree (44, 11%), thus indicating that a neutral response to medical insurance suggests that users lack clarity on whether e-health services are covered or how to claim them. This indicates gaps in awareness, communication, and insurance integration. Addressing these factors could enhance trust and encourage adoption.

This suggests a lack of consensus regarding its impact. In contrast, responses to multilingual support were more distributed. Although the most common response was “Strongly Agree” (81, 21%), indicating a favorable view, a significant number of respondents stated “Disagree” (60, 16%), which reflected mixed perceptions, likely influenced by linguistic diversity or relevance to individual experience. The factor of reduced waiting times received the strongest positive feedback, with 65 (17%) selecting “Agree” and 64 (16%) choosing “Strongly Agree,” showing uncertainty or a lack of direct need among some participants. However, the combined agreement responses were also significant, suggesting the recognition of its importance. Overall, reduced waiting times emerged as the most positive factor, while medical insurance and 24/7 availability drew more ambivalent responses.

Table 3 shows that the ANOVA results indicate that the overall regression model examining the influence of several factors on the new usage of e-health services is statistically significant, with the dependent variable being the new usage of e-health services and the independent variables being the frequency of hospital visits, medical insurance coverage, multilingual support, waiting time reduction, and 24/7 availability. The *p* value (*p* < 0.001) strongly suggests that these predictors collectively have a significant effect on new e-health service usage, since the *p* value is less than 0.05. This means that the model successfully explained a meaningful portion of the variation in users’ adoption of e-health platforms.

Table 4 shows that the Chi-square test was conducted with dependent variables of illness at clinics, recommendations to e-health services, payment processes, trust and partnership with e-health services, and independent variable recommendations of e-health services; the results revealed a strong and statistically significant association between various factors such as illness at clinics, recommendation to e-health services, payment processes, trust and partnership to e-health services, and the recommendation of e-health services. Exposure to illness at clinics showed a high Pearson Chi-square value of 24.028 (*p* < 0.001), indicating that individuals who face health risks in clinical environments are more likely to support and recommend e-health services as safer alternatives. H1 and H2 were accepted because the *p* value was greater than 0.001.

The study revealed several key advantages of e-health consultations, as perceived by users. The most prominent benefit was reduced waiting times, with 42% of respondents strongly agreeing that it was a major factor influencing their switching. Additionally, the availability of services 24/7 and flexibility in scheduling were highly appreciated, especially among the younger and urban populations. Multilingual support was another valued feature, particularly for users from diverse linguistic backgrounds, with 38% expressing a strong agreement on its importance. However, significant challenges were also observed. A large proportion of respondents expressed uncertainty about insurance coverage, with 47% choosing a neutral stance, indicating confusion or lack of awareness. Digital literacy barriers were evident, particularly in rural populations where access to e-health services remains limited. Concerns over trust, privacy, and the accuracy of virtual diagnoses have also been highlighted as key challenges that inhibit broader adoption.

Regarding patient satisfaction, the findings indicated that many users found e-health platforms to be efficient and convenient, especially for follow-ups and non-emergency consultations. Urban and semi-urban residents, particularly those in the 26–35 age group, reported higher satisfaction levels owing to time savings and ease of access. However, trust has become a complex issue. While some respondents appreciated the convenience and accessibility of e-health, trust in the platforms was closely tied to features such as multifactor authentication and visible transparency in service. The Chi-square analysis supported this finding, showing significant associations between prior exposure to illness in traditional clinical settings and increased preference for e-health (*p* < 0.001). This indicates that safety concerns in physical clinics may motivate patients to place greater trust in virtual alternatives when the platforms are secure and transparent.

### 3.1. Strength of the Study

This study has several significant strengths that enhance its value and contribution to ongoing research and practice in digital health adoption. First, it addresses a highly relevant and timely topic, examining the behavioral shift from traditional in-person hospital visits to e-health consultations—an area of major interest globally [9]—especially in the wake of the COVID-19 pandemic. This study captures the transitional behaviors and attitudes of healthcare consumers and providers during a period of major systemic change, making the findings especially pertinent and reflective of current real-world conditions [25]. Another key strength lies in the study’s user-centered focus. Thus, by identifying systemic issues, such as the fragmentation of communication platforms, lack of standardization across e-health services, and challenges of integrating new technologies into traditional care models [26,27].

### 3.2. Weakness of the Study

This study provides important insights into the transition from traditional hospital visits to e-health consultations, and several limitations must be acknowledged. A primary weakness lies in the potential limitations of sample size and representativeness. If the study involved a relatively small number of respondents or participants from a limited geographical area or healthcare setting, the findings may not be generalizable to a broader population. This is especially important given that e-health adoption varies significantly across regions, socioeconomic groups, and age demographics [28,29].

## 4. Discussions

This study provides an in-depth examination of the behavioral shift from traditional in-person hospital consultations to e-health consultations, emphasizing the complex interplay between the individual, technological, and systemic factors driving this change. The implementation of e-health technologies can initially reduce productivity because of the learning curve associated with the new systems.

Our findings align with existing literature that highlights the growing acceptance and integration of digital health solutions. The observed behavioral shift (RQ1) towards e-health showed strong evidence for reducing waiting times and multilingual support (Table 2). The perceived ease of using e-health platforms significantly affects adoption rates. This aligns with the push–pull–mooring (PPM) framework, in which the ease of access and convenience offered by e-health services encourage patients to explore virtual consultations and has proven valuable in explaining why patients choose to shift to e-health platforms [8]. Push factors, such as the discomfort and risks perceived in traditional hospital visits, along with pull factors, such as ease of access and convenience offered by e-health services, encourage patients to explore virtual consultations. Additionally, mooring factors, such as established habits and the level of trust [30] in digital health solutions, play a significant role in shaping this transition. Our results (RQ2) show that perceived benefits such as timesaving and communication support are crucial, which is supported by studies emphasizing perceived usefulness and perceived ease of use as crucial determinants of e-health adoption. Our study (RQ3) implicitly acknowledges challenges such as the inability to conduct physical examinations remotely and difficulty in establishing initial trust and rapport in a virtual setting. Our findings (RQ4) on patient satisfaction and trust have strong statistical associations with trust (multifactor authentication), payment processes, and the recommendation of e-health services (Table 4), reinforcing the critical role of security, transparency, and perceived reliability in fostering patient confidence. This is consistent with studies indicating that higher physician trust and fewer technical difficulties are strongly associated with higher patient satisfaction and trust in telemedicine [31,32,33,34,35,36].

A few limitations of the study were the short time for data collection, and this study lies in the dual nature of technology’s impact on adoption; intuitively, digital health innovations are expected to streamline processes and boost efficiency, yet the study highlights that there is an initial dip in productivity, not just on the provider side but also on the patient side, due to unfamiliarity. This contradicts the common assumption that digital transformation is an immediate upgrade, rather than a transitional process that requires adaptation and training. The push–pull–mooring framework reveals emotional and psychological friction (such as trust and habit) in what is often seen as a technical or logistical shift. Perceived ease of use is more influential than clinical effectiveness in shaping adoption behavior, which underscores the human-centered nature of health tech adoption [8].

It also suggests that behavioural economics and user-centered design are as critical as technological innovation in digital health. Adoption hinges on perceptions, trust, and transition support, and not merely on availability or technical capabilities. Effective policies and strong digital infrastructure are essential to support the shift to e-health by ensuring accessibility, data security, and service standardization. Government initiatives must focus on expanding internet access, digital literacy, and insurance integration. These measures directly address users’ trust, access, and continuity of care. While e-health technologies are designed to improve access and convenience, their initial implementation can reduce productivity owing to the learning curve.

### Future Research Directions

Future research should explore the long-term sustainability of patient satisfaction with e-health in a post-pandemic environment in which patients have more choices, moving beyond the challenges of adoption. Studies could also develop a sophisticated protocol for hybrid care models to optimize modality selection based on clinical needs and patient preferences. Additionally, research is needed to explore effective interventions to ensure equitable access and digital literacy across all demographic groups, particularly older adults and those from lower socioeconomic backgrounds.

## 5. Conclusions

The transition towards e-health consultation modalities is contingent upon a complex interplay of factors, notably the catalytic impact of the pandemic, technological advancements, evolving user acceptance, and internal institutional innovations. A profound understanding of these determinants is imperative for healthcare policymakers, service providers, and technology developers to effectively foster the uptake and optimization of e-health solutions. Strategic imperatives include governmental initiatives for expanded rural internet access, digital literacy promotion, and insurance coverage integration. Public awareness campaigns are necessary to cultivate trust and incentivize adoption.

Healthcare institutions should integrate hybrid care models and invest in comprehensive training. Furthermore, technology developers are tasked with ensuring robust security, privacy, and low-bandwidth functionality in addition to fostering interoperability. Concurrently, a national monitoring framework is crucial for tracking utilization patterns and informing prospective policy development. Future implementations can be implemented as a national policy to ensure universal, affordable, high-speed internet access, particularly in rural and remote areas, backed by dedicated government funding and public–private partnerships, and AI-based solutions for effective clinical decision making. It removes financial barriers for patients and incentivizes providers to offer e-health services. To implement and enforce national technical standards for interoperability between all electronic health record (EHR) systems and e-health platforms, ensuring seamless and secure data exchange.

## Figures and Tables

**Table 1 healthcare-13-01784-t001:** Demographic data.

S No.	Demographic Variable	Description	Frequency	Percentage (%)
1.	Age Group in Years	18–25	125	33
		26–35	142	37
		36–45	34	9
		46–60	40	10
		60 Above	44	11
2.	Gender	Male	148	38
		Female	237	62
3.	Education Level	No formal education	0	0
		High school	111	29
		Bachelor’s degree	205	53
		Master’s degree	69	18
4.	Location	Urban	233	61
		Semi–Urban	75	19
		Rural	77	20

**Table 2 healthcare-13-01784-t002:** Factors influencing the e-health consultation service.

S No.	Factors Influencing	Description	Frequency	Percentage
1.	Medical Insurance	Strongly Disagree	44	11%
		Disagree	50	12%
		Neutral	180	47%
		Agree	70	18%
		Strongly Agree	41	11%
2.	Multilingual Support	Strongly Disagree	37	9%
		Disagree	60	16%
		Neutral	60	16%
		Agree	81	21%
		Strongly Agree	147	38%
3.	Reduce Waiting Times	Strongly Disagree	47	12%
		Disagree	46	12%
		Neutral	65	17%
		Agree	64	17%
		Strongly Agree	163	42%
4.	24/7 Availability	Strongly Disagree	40	10%
		Disagree	44	11%
		Neutral	165	43%
		Agree	88	23%
		Strongly Agree	48	13%

**Table 3 healthcare-13-01784-t003:** ANOVA factors on new usage of e-health services.

Model		Sum of Squares	DF	Mean Square	F	Sig.
1	Regression	14.469	5	2.894	6.999	<0.001
	Residual	156.706	379	0.413		
	Total	171.174	384			
A. Dependent variable: New usage						
B. Predictors: (constant) Frequency of Hospital Visit, Medical Insurance Coverage, Multilingual Support, Waiting Time Reduction, 24/7 Availability						

**Table 4 healthcare-13-01784-t004:** Chi-square—factor recommending e-health services.

S No.	Chi-Square	Value	DF	Asymptotic Significance (2 Sided)
1.	Exposure to illness at Clinics and recommendations of e-health services	24.028	1	<0.001
2.	Certificates and recommendations of e-health services	24.201	1	
3.	Payment process and recommendation of e-health services	15.394	1	
4.	Trust (multifactor authentication) and recommendation of e-health services	26.911	1	
5.	Partnership and recommendation of e-health services	15.048	1	

## Data Availability

The original contributions presented in this study are included in the article. Further inquiries can be directed to the corresponding authors.

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
