# Peer review of "An Analysis of Switching Behavior from Traditional Hospital Visit to E-Health Consultation"

_healthcare, 2025, doi:10.3390/healthcare13151784_

Round 1

Reviewer 1 Report

Comments and Suggestions for Authors

Attached

Comments on the Quality of English Language

The quality of English can be improved.

Author Response

Response to Reviewer 1 Comments

1. Summary

Thank you very much for taking the time to review this manuscript. Please find the detailed responses below and the corresponding revisions in track changes in the resubmitted files.

2. Questions for General Evaluation

Reviewer’s Evaluation

Response and Revisions

Does the introduction provide sufficient background and include all relevant references?

Can be improved  

The background of the study was improved using concepts from the unified theory of acceptance and the technology acceptance model.

Are all the cited references relevant to the research?

Yes/Can be improved/Must be improved/Not applicable

Is the research design appropriate?

Can be improved                                       

Details regarding the questionnaire, Likert scale, and pilot study was briefed in the research design

Are the methods adequately described?

Can be improved                  

The methodology of the study was improved by addressing the limitations, questionnaire, and study design

Are the results clearly presented?

Can be improved

Results were improved by connecting the objectives with results

Are the conclusions supported by the results?

Can be improved

Further conclusion was enhanced by incorporating relevant articles and recommendations for policy makers and administrators

3. Point-by-point response to Comments and Suggestions for Authors

Comments 1:  Introduction

1.     Some paragraphs are repeated (lines 90–129 seem to be duplicates of 31–88). Please remove redundancy

2. Sentence flow could be improved: some sentences are overly long or complex, diluting the message.

3. The introduction can be enhanced by clearly stating the research gap in the final paragraph.

4.     There is inconsistent use of tenses and occasional minor grammatical issues (e.g., “makig,” “hosptial,” “descritpive”).

Response 1:

1. Thank you for pointing this out. We agree with this comment. Therefore, we have reframed the paragraphs in the introduction and discussion phase – page number 1-4 and discussion page no 9

2.     Sentence flow for the overall article has been corrected page no 1- 14

3.     Research gap has been mentioned in the last paragraph of introduction page, no. 3, lines 93-95.

4.     All grammatical corrections were made throughout out the manuscript pages 1-14.

Comments 2: Materials and Methods

1.     The study design lacks details about how convenience sampling was implemented and how bias was minimized.

2. The pilot study with 50 participants is mentioned but not well integrated into the main study (e.g., were the same tools used?).

3. The “Literature Review” (Section 2.6) seems misplaced under Methods. Consider making this a separate section.

4. The Likert scale description should clarify which items used 2-point and which used 5point scales.

5. Grammar and syntax issues persist (e.g., “section A consist” → “consists,” “data enriching the study” is awkward).

Response 2: Agree. We have, accordingly, modified the changes to emphasize this point.

1.     The reason beyond why convenience sampling is chosen was explained in the study design and procedure (subheading 2.2), and how bias was minimized using ethical approval and demographic factors were explained on page 3, lines 112-119

2.     The pilot study details were explained in the study design and procedure (subheading 2.2), page no. 4, first paragraph, lines 126-130.

3.     Literature review was placed as separate section (3), page number 5, lines 177–196.

4.     The explanation for the 5-point Likert scale used to develop questionnaire was mentioned under section 2.3, Questionnaire, line 151-164

5.     Grammar and syntax errors were corrected throughoutthe report from page 1 to 14.

Comments 3:

1.     Interpretation of findings needs more clarity (e.g., why did most people respond neutrally to medical insurance?)

2.     Tables lack precise formatting—consider aligning them neatly and labeling them clearly

3.     Some interpretations overstate significance without caution (e.g., describing a high neutral response as “strongly disagreed” is inaccurate)

4.     Include effect sizes or confidence intervals where possible

5.     More visual elements (e.g., charts or graphs) would enhance clarity and engagement

Response 3: Agree. We have, accordingly, revised the corrections suggested

1.     Findings were explained briefly in the results part—discussion part—page number 9–11.

2.     Tables were aligned properly, and the heading of the tables were changed as per the requirements. Table 1 – page no. 6, table 2 – page no. 6, table 3 – page no. 7

3.     Sentences were reframed in the interpretations page no. 7—first paragraph, third line

4.     Confidence level 95.5% was mentioned in the ANOVA interpretation page, no. 7, last paragraph, lines 240-246

Comments 4:

1.     Content is repeated (lines 279–292 are almost identical to lines 286–291). Consolidate for brevity.

2.     Clarify how the findings support or contradict previous research. The discussion remains mostly descriptive.

3.      Include more critical evaluation—what were the limitations of the statistical tests used? What surprised you?

4.     The role of policy and infrastructure could be elaborated further as practical implications.

Response 4: Agree. We have, accordingly, revised the following points to emphasize the suggestions provided

1. The whole document was proofread, and all the repetitive statements were corrected and highlighted as per the suggestions—page no. 1-14.

2. Previous research work on e-health consultation was mentioned in the discussion part page no. 9, in the discussion section 5

3.     Limitation of sample size was mentioned in the section 4.2 weakness of the study page 9, lines 301-305

4.     The role of policy and infrastructure were mentioned in the discussion section, page number 10, line 349 to 355

Comments 5:

1.     The paragraph lacks depth—it could be enhanced by reflecting on implications for future research or specific health policy reforms.

2.      Language issues such as redundancy (“driven by pandemic-induced necessities, technological advancements, user perceptions, and institutional innovations” could be condensed).

3.     Rephrase the conclusion to avoid generic language and make it more action       oriented.

Response 5: Agree. We have, accordingly, revised the changes to emphasize this point.

1.     The whole document was reframed by including the health policy, future research and limitations. page no 1-14

2.     The document was proof read and certain terminologies were corrected as per the requirement page no 1-14

3.     Section 6 – conclusion was rephrased to avoid the grammar language page no 11 lines 366-386

Comments 6:

1.      Very comprehensive and recent; however, some sources are repeated (e.g., Bhattacherjee, 2001 appears twice).

2.     Verify citation formatting to align with journal guidelines (ensure consistent use of italics, bolding, DOI formatting).

Response 6: Agree. We have, accordingly, modified the provided corrections to emphasize this point.

1.     Reference section was organized in a APA format

2.     Consistent use of reference were made using APA format

7.Key Comments:

1.     Avoid repetition of paragraphs across the introduction and discussion.

2.     Expand on policy recommendations—how can governments, hospitals, or tech developers use this information?

Response 7:

1.   The whole document was proof read and all the repetitive statements were corrected and highlighted as per the suggestions- page no 1-14

2.   Policy recommendations were provided in the discussion part – page no 10, line 349 -355

8. Response to Comments on the Quality of English Language

Point 1: The quality of English can be improved.

Response 1: The whole article was undergone grammar correction and the content was phrased in the academic requirement

5. Additional clarifications

Reviewer 2 Report

Comments and Suggestions for Authors

The abstract is structured; correct grammatical errors and report key statistical findings accurately (test statistic, df, p-value, effect size if available; The significant statistical p-values is p<0,05 as stated in the methodology section, this is in contrast to what is declared in the abstract  - line 23 ;check the keywords in accordance with MeSH.

In the introduction, explicitly state a relevant theoretical framework (e.g., Push-Pull-Mooring - PPM) early in this section to guide the study. The objectives should be stated as research questions (RQs).

The methods are presented in subsections. The questionnaire description is confusing and lacks rigor; clearly define the key constructs/variables measured in each section; specify the exact scales used etc. Objectives 3 and 4 (advantages/challenges, satisfaction/trust) are not clearly linked to specific analyses in the results section

In the results section there is a major omission - Core results related to the primary objectives (behavioral shift, factors influencing adoption, advantages/challenges, satisfaction/trust vs in-person) are missing; where are the main findings answering objectives 1-4? The abstract mentions hypothesis testing, but no specific hypotheses are stated in the methods, and no hypothesis results are reported in the results section. p-values are mentioned but often without the corresponding test.

The discussion is extremely short in contrast to the introduction section, some of the information provided in the introduction should be moved to this section and the results better interpreted. Also, please give precise examples of remote care practices in the healthcare field, by referring to scientific literature (for e.g. doi: 10.3390/healthcare13070736 ). Future research directions should be stated at the end.

The conclusion section is very brief; succinctly summarize the key findings specific to this study.

The references are adequate but should be extended as suggested above.

Editing recommendations: the references should be noted with numbers in text under “[ ]”; the tables should be edited in APA academic style; all tables are misplaced, they should be inserted after they are cited in text not before.

Author Response

Response to Reviewer 2 Comments

1. Summary

Thank you very much for taking the time to review this manuscript. We appreciate your insightful comments and suggest, which have significantly helped us to improve our article in terms of clarity, rigor and completeness. Please find our detailed responses below and the corresponding revisions highlighted in red within the re-submitted files.

2. Questions for General Evaluation

Reviewer’s Evaluation

Response and Revisions

Does the introduction provide sufficient background and include all relevant references?

Can be improved  

Background of the study was improved using concepts unified theory of acceptance and Technology Acceptance Model

Are all the cited references relevant to the research?

Yes/Can be improved/Must be improved/Not applicable

Yes

Is the research design appropriate?

Yes                                       

-

Are the methods adequately described?

Must be improved                  

Methodology of the study was improved by addressing the limitations, questionnaire and study design

Are the results clearly presented?

Must be improved

Results were improved by connecting the objectives with results

Are the conclusions supported by the results?

Can be improved

Further conclusion was enhanced by incorporating relevant articles and recommendations for policy makers, administrators

3. Point-by-point response to Comments and Suggestions for Authors

Comments 1:  The abstract is structured; correct grammatical errors and report key statistical findings accurately (test statistic, df, p-value, effect size if available; The significant statistical p-values is p<0,05 as stated in the methodology section, this is in contrast to what is declared in the abstract  - line 23 ; check the keywords in accordance with MeSH.

Response 1: Thank you for pointing this out. We agree with this comment.

1.     Therefore, we have corrected the grammatical errors.

2.     The contrast of p-values in the abstract is corrected in line 23-page number 1

3.     Reviewed and updated the keywords in accordance with MeSH (Page 1, line 29)

Comments 2: In the introduction, explicitly state a relevant theoretical framework (e.g., Push-Pull-Mooring - PPM) early in this section to guide the study. The objectives should be stated as research questions (RQs)

Response 2:

1.     Agree. We have, accordingly, modified the changes to emphasize this point.

2.     Theoretical framework was explained in the manuscript in the introduction (line 79 and Page number 2) and in discussion (line 317, page 9, second paragraph in the discussion).

3.     We have also rephrased the study objectives into clear research questions (RQs) to enhance the study’s focus and alignment (Line 97 – 102, Page number 3) in the revised manuscript.

Comments 3: The methods are presented in subsections. The questionnaire description is confusing and lacks rigor; clearly define the key constructs/variables measured in each section; specify the exact scales used etc. Objectives 3 and 4 (advantages/challenges, satisfaction/trust) are not clearly linked to specific analyses in the results section

Response 3:

1.     Agree. We have, accordingly, revised the key variables in the questionnaire session to emphasize this point. Likert 5-point scale was used to develop the questionnaire (line 151-164 and page number 4)

2.     Objectives 3 and 4 were linked with the result session as suggested (line 264 – 286, page number 8 & 9)

Comments 4: In the results section there is a major omission - Core results related to the primary objectives (behavioral shift, factors influencing adoption, advantages/challenges, satisfaction/trust vs in-person) are missing; where are the main findings answering objectives 1-4? The abstract mentions hypothesis testing, but no specific hypotheses are stated in the methods, and no hypothesis results are reported in the results section. p-values are mentioned but often without the corresponding test.

Response 3: Agree. We have, accordingly, revised to emphasize this point.

1.     Based on the objectives, results and findings are addressed. (line 250 – 286, page number 8

2.     Hypothesis for the study is stated and reported in the results section. (line 240 – 246, page number 7; 250 – 259, page number 8)

3.     We have ensured that all p-values are reported with their corresponding test statistics [F-values for ANOVA, Chi-square values and degrees of freedom(df)] providing a more rigorous presentation of the statistical findings.

Comments 4: The discussion is extremely short in contrast to the introduction section, some of the information provided in the introduction should be moved to this section and the results better interpreted. Also, please give precise examples of remote care practices in the healthcare field, by referring to scientific literature (for e.g. doi: 10.3390/healthcare13070736 ). Future research directions should be stated at the end.

Response 2: Agree. We have, accordingly, revised to emphasize this point.

We have thoroughly revised this section by directly linking the results to the research questions and hypotheses.

1.     Incorporating relevant scientific literature, including the suggested DoI: 10.3390/healthcare13070736 and the other studies from the research advantages/ challenges and patient satisfaction/ trust to contextualize our findings within the broader academic discourse. (line 313 – 346; page number 10)

2.     Moving some general background and contextual information from the introduction to the discussion.

3.     Adding a dedicated subsection for future research directions (4.1) at the end of the discussion. (Page number 10; line 357 – 363)

Comments 4: Editing recommendations: the references should be noted with numbers in text under “[ ]”; the tables should be edited in APA academic style; all tables are misplaced, they should be inserted after they are cited in text not before

Response 2: Agree. We have, accordingly, revised to emphasize this point.

1.     Corrected the references as per the suggestions in the entire manuscript.

2.     Tables are placed appropriately after the citations in the manuscript. (Line 214, 238, 248, 262; page number 6, 7 & 8)

4. Response to Comments on the Quality of English Language

Point 1: The English is fine and does not require any improvement.

Response 1:

The whole article was undergone grammar correction, and the content was phrased in the academic requirement

5. Additional clarifications

Reviewer 3 Report

Comments and Suggestions for Authors

 1. The manuscript has frequent grammatical errors, awkward phrasing, and inconsistent terminology (e.g., “convience sampling,” “frequenc,” “be e health consultation,” “makig,” etc.).  I recommend professional language editing to ensure polished academic writing throughout the manuscript. Ensure consistency in key terms: standardize “e-health consultation” or “telehealth” instead of alternating terms.
 2. The authors must clarify the sampling method (convenience sampling is mentioned, but no justification is provided).
 3. How the questionnaire was developed and validated.
 4.  How was the sample size (n=385) determined for statistical adequacy?
 5. Integrate theoretical models (e.g., Technology Acceptance Model, Unified Theory of Acceptance and Use of Technology) more deeply to provide conceptual grounding.
 Include more recent studies (2023–2025) that address post-pandemic e-health adoption to increase relevance.
6.  The authors should explain the statistical analysis and clearly define dependent and independent variables for ANOVA and Chi-square tests. They should report actual effect sizes, confidence intervals, and assumptions testing (e.g., homogeneity of variance for ANOVA). Tables should include clearer titles and concise interpretations under each table.
 7.  The authors should discuss the limitations, sample bias (urban and well-educated respondents are overrepresented).  Potential response bias from online data collection (Google Forms). The limitation of self-reported behavior vs. actual behavior.
 8. The authors should offer more concrete recommendations for policymakers (e.g., bridging digital divides, infrastructure improvements) and healthcare providers (e.g., training for both patients and staff). They should also suggest specific areas for future research (e.g., longitudinal studies, rural populations, AI-driven telehealth).
 9. Ensure proper section numbering (some sections seem merged or repetitive).   Check journal-specific guidelines for formatting references, abstract length, and keywords. Remove redundant text (e.g., repeated sentences about Push–Pull–Mooring framework).

Author Response

Response to Reviewer 3 Comments

1. Summary

Thank you very much for taking the time to review this manuscript. Please find the detailed responses below and the corresponding revisions in track changes in the re-submitted files.

2. Questions for General Evaluation

Reviewer’s Evaluation

Response and Revisions

Does the introduction provide sufficient background and include all relevant references?

Can be improved  

Background of the study was improved using concepts unified theory of acceptance and Technology Acceptance Model

Are all the cited references relevant to the research?

Yes/Can be improved/Must be improved/Not applicable

Is the research design appropriate?

Can be improved                                       

Details regarding the questionnaire, Likert scale and pilot study was briefed in the research design

Are the methods adequately described?

Can be improved                  

Methodology of the study was improved by addressing the limitations, questionnaire and study design

Are the results clearly presented?

Can be improved

Results were improved by connecting the objectives with results

Are the conclusions supported by the results?

Can be improved

Further conclusion was enhanced by incorporating relevant articles and recommendations for policy makers, administrators

3. Point-by-point response to Comments and Suggestions for Authors

Comments 1:  

1.     The manuscript has frequent grammatical errors, awkward phrasing, and inconsistent terminology (e.g., “convience sampling,” “frequenc,” “be e health consultation,” “makig,” etc.).  I recommend professional language editing to ensure polished academic writing throughout the manuscript. Ensure consistency in key terms: standardize “e-health consultation” or “telehealth” instead of alternating terms.

Response 1:

1.     Thank you for pointing this out. We agree with this comment. Therefore, we have reframed the paragraphs from the introduction till discussion phase – page number 1- 14

All grammatical corrections were made through out the manuscript page 1- 14

Comments 2:

The authors must clarify the sampling method (convenience sampling is mentioned, but no justification is provided).

Response 2: Agree. We have, accordingly, modified the changes to emphasize this point.

1.     The reason beyond why convenience sampling is chosen was explained in the study design and procedure (sub heading 2.2) and how bias was minimized using ethical approval, demographic factors were explained in the study page no – 3; line 115-119

Comments 3:

 How the questionnaire was developed and validated?

Response 3:  Agree. We have, accordingly, revised the corrections suggested

1.     The development of questionnaire was explained in the research design section 2.3 and pilot test was used to validate the questionnaire

Comments 4:

 How was the sample size (n=385) determined for statistical adequacy?

Response 4: Agree. We have, accordingly, revised the following points to emphasize the suggestions provided

1.     Based on the population of the healthcare professionals the sample size has been fixed and details of the same is explained in the section 2.1 participants

Comments 5:

1.     Integrate theoretical models (e.g., Technology Acceptance Model, Unified Theory of Acceptance and Use of Technology) more deeply to provide conceptual grounding.

2.      Include more recent studies (2023–2025) that address post-pandemic e-health adoption to increase relevance.

Response 5: Agree. We have, accordingly, revised the changes to emphasize this point.

1.     The models like technology acceptance model and unified theory of acceptance were explained in the introduction page no 2 line 42-46.

2.     Recent studies conducted 2023-2025 were included in the literature review section 3. Lines 190 -196

Comments 6:

 The authors should explain the statistical analysis and clearly define dependent and independent variables for ANOVA and Chi-square tests. They should report actual effect sizes, confidence intervals, and assumptions testing (e.g., homogeneity of variance for ANOVA). Tables should include clearer titles and concise interpretations under each table.

Response 6: Agree. We have, accordingly, modified the provided corrections to emphasize this point.

1.     The analysis part was brief explained in results section where the dependent and independent variables were mentioned and table was reframed with required information section 2.4 statistical analysis page no 5; line 169 - 171

 Comments 7:

The authors should discuss the limitations, sample bias (urban and well-educated respondents are overrepresented).  Potential response bias from online data collection (Google Forms). The limitation of self-reported behavior vs. actual behavior.

Response 7:

1.    Limitations and sample bias were explained in the weakness section 4.2 and discussion part section 5 and the specific reason for the limitations were explained in the discussion section 5

Comments 8:

The authors should offer more concrete recommendations for policymakers (e.g., bridging digital divides, infrastructure improvements) and healthcare providers (e.g., training for both patients and staff). They should also suggest specific areas for future research (e.g., longitudinal studies, rural populations, AI-driven telehealth).

Response 8:

Specific recommendations and future study expansion was explained in the discussion part section 5.1 page no 10; line 358 - 364

Comments 9:

 Ensure proper section numbering (some sections seem merged or repetitive). Check journal-specific guidelines for formatting references, abstract length, and keywords. Remove redundant text (e.g., repeated sentences about Push–Pull–Mooring framework).

Response 9:

 The whole document was proofread and formatted as per the journal guidelines and all the repeated sentences were removed, page no 1- 14

10. Response to Comments on the Quality of English Language

Point 1: The English could be improved to more clearly express the research.

Response 1: The whole article was undergone grammar correction and the content was phrased in the academic requirement

11. Additional clarifications

Round 2

Reviewer 1 Report

Comments and Suggestions for Authors

None

Author Response

We are writing to express our sincere gratitude for the decision to accept our manuscript "healthcare-3711563."

We would also like to extend our heartfelt thanks to the reviewer for their time and effort for the meticulous and constructive review.

Reviewer 2 Report

Comments and Suggestions for Authors

The authors have respected most of the initial indications (please respect them all). Some adjustments can still be made: please do not use references in the conclusion section; move Section 3 ("Literature Review") as first subsections of "Materials and Methods" to better ground hypotheses.

Author Response

We are writing to express our sincere gratitude for reviewing our manuscript "healthcare-3711563." 

3. Point-by-point response to Comments and Suggestions for Authors

Comment 1: Please do not use references in the conclusion section;

Response 1: Reference in the conclusion section were removed (Page 10 & 11)

Comment 2: Move Section 3 ("Literature Review") as the first subsection of "Materials and Methods" to better ground hypotheses.

Response 2: Literature review has been moved to “materials and methods” (page no. 3, lines 104–124).

We would like to extend our heartfelt thanks to the reviewer for their time and effort in conducting such a meticulous and constructive review. We truly appreciate the thoroughness of their feedback.

Thank you for your consideration and for managing the review process.

Reviewer 3 Report

Comments and Suggestions for Authors

The authors have already revised all of the reviewer's issues. Thank you for carefully revising point-by-point and improving your research. 

Author Response

We are writing to express our sincere gratitude for the decision to accept our manuscript "healthcare-3711563."

We would also like to extend our heartfelt thanks to the reviewer for their time and effort for their meticulous and constructive review.